# Higher Uncertainty Leads to Less Exploration in a Combinatorial Discovery Game

## Abstract

How do people decide whether it is worth pursuing innovation? For example, in machine learning new methods often result from combining existing methods, but there is a risk that a given combination will not work. While seasoned experts could use their intuitions gained through experience to decide whether some combinations are worth trying out, novices to the field have to learn these insights while trying to maximize their rewards. Here, we formalize this problem and derive optimal policies for agents who know, or do not know, how likely each kind of combination is to succeed, emulating the effects of expert knowledge. Our model predicts that novices should not only gather fewer rewards, but also explore systematically less than the experts. An online behavioral experiment ($n = 300$) supports this finding, showcasing the profound impact of domain expertise in guiding innovative decision making in a combinatorial space.

## 1 Introduction

People often create new things by recombining (parts of) existing things [1]. To name a few examples, adding engines to machines unleashed an era of industrial revolution; combining artificial neural networks with internet-scale data produced astonishing artificial intelligence systems. More recently, pooling together a range of pipelines created AI scientists that can automatically propose ideas, test hypotheses, and publish papers [2]. In short, a large part of human discovery has a fundamentally combinatorial nature.

The space of combinatorial discoveries can have structure, and knowing the odds of attempting recombination plays a vital role in the decision-making process. Take medical research for example: experts with good knowledge of whether certain chemicals and proteins go together can make more efficient experiment designs than those who have to figure out these information. [3] showed that, when the success rates of recombination are known and the horizon is finite, this decision problem can be expressed as a Markov decision process and hence solved. In a behavioral experiment, [3] found that people's behaviors are in line with those solutions.

However, in more realistic settings, the success rate is not always known. Here, we examine the influence of such domain-level uncertainty. We ask what a rational agent should do to maximize their rewards when the success rate of attempting recombination is unavailable. As the number of unknown domains changes, would a rational agent react differently? How much gain does knowledge of success rates bring us? We answer these questions both from a Bayesian modeling perspective, and using data collected via an online behavioral experiment. Understanding these questions crystallizes the dynamics of how people pursue innovation in fields involving combinatorial discoveries, and helps us make better plans to facilitate future discoveries.

Submitted to Workshop on Bayesian Decision-making and Uncertainty, 38th Conference on Neural Information Processing Systems (BDU at NeurIPS 2024). Do not distribute.

## 2 Methods

### 2.1 Combinatorial Discovery Games

Following [3], we first define a basic combinatorial discovery game $\mathbf{G}$ as a tuple $\langle M, T, A, R \rangle$, where $M$ is a set of items, $T$ the game tree containing successful combinations, $A$ a set of actions, and $R$ the reward function. Combining an item $m \in M$ and item $n \in M$ is denoted as $c(m,n)$. If $c(m,n) \in T$, this is a successful combination and will produce a new item, say, $c(m,n) \Rightarrow o$; if $c(m,n) \notin T$, the combination fails and no new item is discovered. Items that cannot be produced by combining other items are base items, $m^0$, with base reward $r$. Climbing up the game tree increases levels: for a combination $c(m^i, m^j) \Rightarrow m^k$, $k = \max(i,j) + 1$, and the reward associated with item $m^k$ is $w^k \cdot r$. Players can take two actions $A = \{use, combine\}$. Action $use$ receives the rewards associated with the item, $R(use(m^k)) = w^k r$, and action $combine$ combines two items of the player's choice. The immediate reward for taking this action is always zero, $R(combine(m,n)) = 0$.

A discovery game can be parameterized by a success rate $p \in [0,1]$—the probability of receiving a successful discovery for any given item, and the reward increase rate $w > 1$. [3] showed that these definitions form a Markov decision process (MDP), and under a finite horizon of $D$ steps in total, the optimal policy in this particular setting corresponds to an optimal stopping problem [4]: One should keep attempting innovation ($combine$) until a switch point $d$, then focus on collecting the highest possible existing rewards ($use$). The expected return for switching at step $d$ is

$$\mathbb{E}_{\pi(d)} = (n - d) \left( \sum_{i=0}^{d} \binom{d}{i} (pw)^i (1-p)^{d-i} \right) r \tag{1}$$

and the optimal switch point is $d^* = \arg\max_d \mathbb{E}_{\pi(d)}$. Solving Equation 1 analytically states that a rational player should switch from $use$ to $combine$ when there is $d'$ steps left, where

$$d' \geq \frac{1}{p(w-1)} + 1. \tag{2}$$

### 2.2 Domain Uncertainty

The above policy specifies what a rational agent should do when they have complete information about the success rates and reward increase rates. In the real world, this setup corresponds to an expert with strong domain knowledge, enabling informed decisions that drive innovation and yield desirable outcomes.

To account for uncertainty in domain knowledge, we extend the setup in [3] to include types of items, aiming to reflect a more realistic scenario where different types of things may be associated with different success rates and reward increase rates—like which kinds of chemicals can be productively combined together. A discovery game with types $\mathbf{G}_\tau$ extends a discovery game $\mathbf{G} \cup \{Z, \sigma, \tau\}$, where type indices $Z = \{1, 2, \ldots, z\}$ is a finite set of integers, and $\sigma : M \to Z$ maps each game item to a type index. For $|Z| = z$ number of types, choosing two types from $z$ with replacement—allowing same-type combination—is given by $\binom{z+2-1}{2} = \frac{(z+1)!}{2!(z-1)!}$ ways of combining types of items. We denote each kind of these combinations by $\tau$.

Each kind of combination has its own success rate $p_\tau$ and reward increase rate $w_\tau$. Experts are defined as those knowing the $p_\tau$ and $w_\tau$ for all $\tau$. An expert can thus estimate the maximum total reward for each kind of combination using Equation 1, and apply Equation 2 to compute the optimal switch point $d_\tau^*$ for the kind of combination that produces the highest rewards. Formally, the rational decision is to optimize along the most rewarding $\tau$:

$$\hat{\tau} = \arg\max \mathbb{E}_{\pi(d_\tau^*)}. \tag{3}$$

Non-experts, or novices, are not blessed with such information. They are tasked with both inferring the relevant parameter values, and maximizing their rewards. While the reward increase rate $w$ can be known immediately after a successful combination is found, the success rate $p$ poses a harder inference problem. Without loss of generality, we model a novice's belief about the success rate with a beta distribution

$$p_\tau \sim \text{Beta}(\alpha_\tau, \beta_\tau). \tag{4}$$

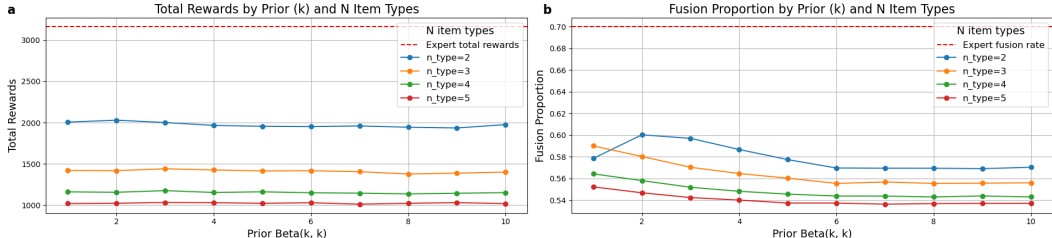

Figure 1: Comparing the performance of a novice learning via Bayesian inference and an expert. **a.** max total rewards achieved by the novice per prior and number of unique item types. **b.** fusion proportions. Red dotted lines mark expert's performances.

Given this prior, the novice player updates their belief about $p_\tau$ after observing $x_\tau$ successes out of $n_\tau$ attempts of *combine* actions for this kind of combination, with

$$p_\tau \mid x_\tau, n_\tau \sim \text{Beta}(\alpha_\tau + x_\tau, \beta_\tau + n_\tau - x_\tau). \tag{5}$$

The novice player can integrate the belief defined in Equation 5 with Equation 2 to estimate a switch point $\tilde{d}_\tau^*$ given their current belief of success rate $p_\tau$ for $\tau$, and it is possible to compute the max expected reward for each $\tau$ using Equation 1. The novice player can then apply the same decision rule in Equation 3 to choose which kind of combination they will interact with, and choose the action

$$a_\tau = \begin{cases} combine_\tau & \text{if } n_\tau < \tilde{d}_\tau^*, \\ use_\tau & \text{otherwise.} \end{cases} \tag{6}$$

Note that Equations 1–6 effectively equate to Thompson Sampling [5], a posterior sampling algorithm widely used in online learning problems [6].

For a novice's prior belief, we set $\alpha_\tau = \beta_\tau = k$, representing no biases about whether a kind of combination is particularly promising or devastating. We ran simulations with symmetric beta priors $k = 1, \ldots, 10$, and number of item types $z = 2, 3, 4, 5$, leading to $|\tau| = 3, 6, 10, 15$ respectively. Figure 1 summarizes these simulation results, and reveals that novices both gain fewer rewards and attempt fusion less frequently than the experts. How strong the prior is—reflected by the $k$ values— has no substantial influence on the learner's behavior; domain richness—measured by the number of item types—has a much stronger impact: The richer the domain is, the bigger the differences between the novice's and the expert's performances are. These results suggest that higher uncertainty for the novices can lead to more conservative behavior in this particular setting.

## 3 Experiment

We test these model predictions in a pre-registered online behavioral experiment (`https://osf.io/x2ymd/?view_only=84c4b13ea553457c8661e426169623a9`). The experiment was approved by the university ethics committee (ref. no. omitted for anonymity).

Three-hundred participants were recruited from Prolific Academic (age $35 \pm 12$, 55% female). The experiment took $10 \pm 19$ minutes. No participants were excluded from analysis. All participants gave informed consent before undertaking the experiment.

### 3.1 Design

We used a cover story of making alien crystals that could be used to generate points. Participants could either use an existing crystal (action *use*), or fuse any two existing crystals together (action *combine*). A fusion attempt led to a new crystal with probability $p$. Participants could take 10 actions in each game. Each game had 4 types of alien crystals indicated by 4 shapes: square, triangle, diamond (upside-down triangle), and circle. These four shapes form 10 kinds of combinations (square + square, square + circle, etc.). All crystals start at the base level with $r = 100$ points. We set the reward increase rate $w = 1.5$ for all kinds of combinations and let all participants know this was the case. Notably, one of the ten kinds of combinations had a high success rate ($p_H = 0.8$),

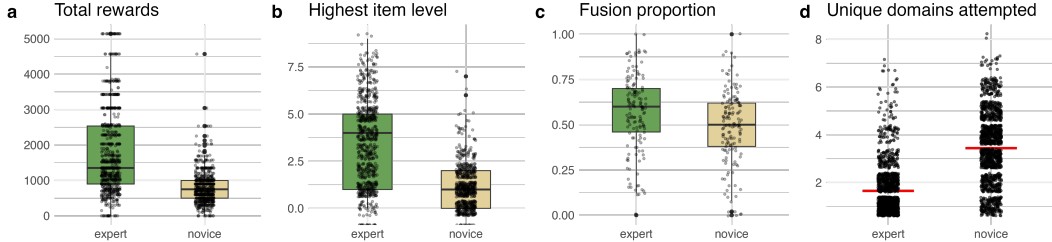

Figure 2: Behavioral results. Each dot is an individual data point. Central lines in the boxes are median values; the lower and upper edges of the boxes are the first and third quantiles. Red lines in sub-figure **d** are mean values.

while the others had lower success rates ($p_L = 0.2$). Participants were randomly assigned to one of the two conditions. For those in the *expert* condition, they were explicitly told the success rate parameters. Participants assigned to the *novice* condition were only told the $p_H$ and $p_L$ values, but had no information of which kind of combination was associated with $p_H$ or $p_L$.

Participants first read the instructions and had to pass a comprehension quiz to start the task. Each participant completed 5 practice trials and then 5 task trials. Each trial has only one high-$p$ kind of combination, sampled randomly and independently. The experiment ended with a short debriefing. See the experiment in action at `https://bz.velezlab.opalstacked.com/crystals-ep/p/exp.html`.

## 3.2 Selected Results

We only analyzed data in the task trials. As illustrated in Figure 2a-b, overall, participants in the *expert* condition collected more total rewards and created items with higher levels. We conducted a mixed-design Analysis of Variance (ANOVA) with condition as the primary factor and task as a repeated measure to assess the effects. The results reveal that for total rewards, there is a significant effect of condition ($F(1,298) = 163.479$, $p < 0.0001$), but not for task ($F(3.81,1136.06) = 1.522, p = 0.196$) or their interaction ($F(3.81,1136.06) = 0.374, p = 0.819$); similarly for item levels, condition has a significant effect ($F(1,298) = 179.926, p < 0.0001$), but not for task ($F(3.88,1157) = 2.114, p = 0.079$) or the interaction between condition and task ($F(3.88,1157) = 1.264, p = 0.283$).

Crucially, as predicted by the model, participants in the *novice* condition attempted fewer fusion actions (Figure 2c). A mixed-design ANOVA with condition as the primary factor and task as a repeated measure indicated a significant effect ($F(1,298) = 13.183$, $p = 0.000332$), and not for task ($F(3.75,1116.59) = 0.285$, $p = 0.877$) or the interaction between condition and task ($F(3.75,1116.59) = 0.928, p = 0.442$).

Interestingly, despite being less exploratory in the sense of attempting fewer fusions, participants in the *novice* condition interacted with more item types (Figure 2d). Again, a mixed-design ANOVA with condition as the primary factor and task as a repeated measure indicated a highly significant effect of condition ($F(1,268) = 329.443, p < 0.0001$), and not for task ($F(3.83,1027.65) = 0.770, p = 0.540$) or the interaction ($F(3.83,1027.65) = 0.370, p = 0.822$).

## 4 Conclusion

Advancing discoveries via recombination is a crucial aspect of human intelligence, ranging from creating physical tools to proposing new theories, or even literally creating novel chemical compounds or protein structures. We examined decision policies in a task sharing a similar combinatorial nature, and in particular investigated the influence of domain uncertainty. As both predicted by the model and supported by empirical data, higher uncertainty not only led to fewer rewards or less advanced items, but also lower rate of exploration, although touching on a wilder range of domains. These results emphasize the importance of integrating domain-expertise in guiding effective exploration, and warn the undesirable possibilities of reward-driven policies in highly uncertain domains.

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
