# OpenReview forum: "Higher Uncertainty Leads to Less Exploration in a Combinatorial Discovery Game"
_NeurIPS.cc/2024/Workshop/BDU — NeurIPS BDU Workshop 2024 Poster_

### Official Review · Reviewer_HCsN · 2024-09-19
**Interesting finding**

**Rating:** 6
**Confidence:** 3

**Review:**

## Summary:

The authors follow the assumption that creating new things by recombining existing things (such as drug design by recombining molecular structures) can be formalized by a Combinatorial Discovery Game (CDG). Then, this paper try to answer how domain knowledge affect a rational agent's decision making in a CDG. This work extends existing models to better predict human decision-making behavior at different levels of domain knowledge. Their model shows that novices should not only gather fewer rewards, but also explore systematically less than the experts.  The authors also conducts an online behavioral experiment to support the finding.

In my opinion, the authors' finding is interesting and counter-intuitive (i.e., novices explore less than experts). I believe this finding is useful to help researchers design new exploration methods for reinforcement learning agents (such as knowledge-guided exploration). However, I have some concerns about the technical contributions of this paper, as the extensions to previous models seem to be trivially.



### Clarity :
   The paper is well-structured and the writing is fluent and easy to read.

### Quality :
  The problem focused by this paper is interesting, but the technical contribution seems insufficient. However, It should be emphasized that the content and finding of this paper is valuable and inspiring for me.

### Originality :
   To the best of my knowledge, this work is novel.

### Significance :
   While the finding of this paper is interesting, I am concerned about its significance given the simplicity of the setting and the lack of deeper analysis provided.

---

### Official Review · Reviewer_hg1t · 2024-09-29

**Rating:** 6
**Confidence:** 4

**Review:**

Strengths:

1. Relevance and Timeliness: The paper addresses an important and timely topic in AI and decision-making under uncertainty, which is crucial for understanding innovation processes.

2. Theoretical Foundation: The authors provide a solid theoretical framework, extending previous work on combinatorial discovery games to include domain uncertainty. The use of Bayesian modeling and Thompson Sampling is appropriate and well-justified.

3. Empirical Support: The inclusion of a behavioral experiment with a substantial sample size (n=300) adds empirical weight to the theoretical predictions.

4. Clear Methodology: The paper outlines a clear and replicable methodology, both for the theoretical model and the experimental design.

5. Interdisciplinary Approach: The study bridges theoretical computer science, decision theory, and behavioral psychology, which is valuable for a top AI venue.

Weaknesses:

1. Limited Scope: The study focuses on a specific type of combinatorial discovery game, which may limit its generalizability to real-world innovation processes.

2. Simplification of Expertise: The binary categorization of "expert" vs. "novice" may oversimplify the spectrum of domain knowledge in real-world scenarios.

3. Lack of Deeper Analysis: While the results support the main hypothesis, there's limited discussion on why novices interact with more item types despite being less exploratory overall. This unexpected finding warrants more in-depth analysis.

4. Missing Long-term Implications: The paper doesn't adequately address how the observed short-term behaviors might impact long-term innovation outcomes.

5. Limited Discussion of Practical Applications: The authors could have elaborated more on how these findings might be applied to real-world AI development or innovation strategies.

6. Potential Confounds: The experiment's cover story (alien crystals) might introduce unintended biases or misunderstandings among participants. A more neutral framing could have been considered.

Recommendations:

1. Expand the discussion on the unexpected finding of novices interacting with more item types. This could lead to valuable insights about different exploration strategies.

2. Consider running additional experiments with varying levels of expertise to create a more nuanced understanding of how domain knowledge impacts exploration.

3. Provide more discussion on how these findings might be applied in real-world AI development or innovation processes.

4. Conduct sensitivity analyses to test the robustness of the model under different parameters and assumptions.

5. Include a section on limitations and future work to address potential criticisms and outline next steps for this line of research.

Overall Evaluation:
This paper presents a novel and important contribution to understanding decision-making under uncertainty in combinatorial discovery tasks. The combination of theoretical modeling and empirical testing is a strength. However, the limited scope and some missed opportunities for deeper analysis slightly diminish its impact. With the suggested improvements, this paper could be a strong candidate for acceptance at a top AI venue. As it stands, it's a good fit for a workshop and with revisions could be suitable for a main conference.

---

### Decision · Program_Chairs · 2024-10-09

Accept (Poster)